# Behavior of Piglets in an Observation Arena before and after Surgical Castration with Local Anesthesia

**DOI:** 10.3390/ani13030529

**Published:** 2023-02-02

**Authors:** Regina Miller, Andrea Grott, Dorian Patzkéwitsch, Dorothea Döring, Nora Abendschön, Pauline Deffner, Judith Reiser, Mathias Ritzmann, Anna M. Saller, Paul Schmidt, Steffanie Senf, Julia Werner, Christine Baumgartner, Susanne Zöls, Michael Erhard, Shana Bergmann

**Affiliations:** 1Department of Veterinary Science, Chair of Animal Welfare, Ethology, Animal Hygiene and Animal Husbandry, Faculty of Veterinary Medicine, Ludwig Maximilian University of Munich, LMU Munich, 80539 Munich, Germany; 2Clinic for Swine, Ludwig Maximilian University of Munich, 85764 Oberschleißheim, Germany; 3Center for Preclinical Research, Technical University of Munich, 81675 Munich, Germany; 4Statistical Consulting for Science and Research, Große Seestr. 8, 13086 Berlin, Germany

**Keywords:** animal welfare, pain relief, procaine, lidocaine, mepivacaine, bupivacaine

## Abstract

**Simple Summary:**

Surgical castration of piglets is generally recognized as a painful procedure. Thus, for animal welfare reasons, the German Animal Welfare Act stipulates the use of effective anesthesia during castration. However, whether local anesthesia provides adequate analgesia has been an ongoing debate in Germany. In the present study, we compared the behavior of 178 piglets allocated to various test groups in an observation arena before any of the applied procedures, after administration of the local anesthetic, and 0, 2 and 24 h after surgical castration. The local anesthetic and the injection techniques were evaluated and optimized in three sequential study parts. Overall, the results revealed that when local anesthesia was used, piglets less frequently showed pain-associated behaviors—such as changes in tail position—than piglets of the control group that had been castrated without local anesthesia. Non-castrated piglets showed the fewest pain-associated behaviors in the observation arena. In several test groups, the piglets showed changes in tail position, tail wagging, or hunched-back posture on the day following the procedure. These behaviors differed significantly from those shown before the procedure. The administration of local anesthetics in the present study considerably reduced castration-related pain. However, because local anesthesia has a limited duration of effect, adverse effects due to castration-related pain were still observable one day after castration.

**Abstract:**

Surgical castration of piglets is generally recognized as a painful procedure, but there is currently no gold standard for the assessment of pain behavior in piglets. However, pain assessment is essential for evaluating the effectiveness of local anesthetics. In this study, we investigated the efficacy of four local anesthetics in terms of pain relief during and after surgical castration in three sequential study parts. To do so, we filmed 178 piglets before the applied procedures, after injection of the local anesthetic, and up to 24 h after castration (five observation times in total) in an observation arena and compared their behavior before and after castration and between treatments and control groups. The results showed significant differences in the behavior of the piglets before and after castration and between the sham-castrated control group and the control group castrated without anesthesia. The different local anesthesia treatment groups showed diverging differences to the control groups. The most frequently shown pain-associated behaviors of the piglets were changes in tail position and hunched back posture. We observed a reduction but no complete elimination of the expressed pain-associated behaviors after local anesthesia. Several behavioral changes—such as changes in tail position, hunched back posture or tail wagging—persisted until the day after castration. Owing to the limited duration of the effects of the local anesthetics, local anesthesia did not influence long-term pain.

## 1. Introduction

The International Association for the Study of Pain defines pain as an “unpleasant sensory and emotional experience associated with actual or potential tissue damage or described in terms of such damage” [1]. This definition is the same for all animal species and for humans, although the intensity and expression of pain can differ between species and between individuals of the same species, thus making an objective assessment difficult. Regarding pigs, even experts have difficulties in reliably recognizing pain [2]. However, proper assessment and elimination of pain are mandatory to comply with the requirements of the German Animal Welfare Act and to ensure that the wellbeing of the animals is not negatively affected. According to Article 1 Section 1, Clause 2, no one may inflict unjustified pain, suffering or injury on animals [3]. Therefore, as of 1 January 2021, castration for the purpose of preventing boar taint is only permitted with effective anesthesia in Germany. This regulation applies to about 20 million male piglets annually [4]. Because alternatives—such as boar fattening or immunocastration—do not yet prevail over surgical castration in Germany [5], evidence-based suitable methods of anesthesia that reliably reduce castration-induced pain must be applied. Currently, castration with local anesthesia is not allowed in Germany because its effectiveness in eliminating pain is still a matter of debate.

Whereas humans are mostly able to evaluate and verbally communicate the intensity and quality of pain, veterinary medicine must rely on nonverbal pain parameters [6]. Options for pain assessment in animals include the measurement of physiological parameters such as heart rate, blood pressure or various blood parameters (e.g., catecholamines or cortisol). Because the measurement of these parameters usually requires handling of the animals, the results may be confounded by stress, fear or external factors [7]. Especially for castration-induced pain, but also for other pain-inducing situations, behavior observations were shown to provide reliable assessment parameters [7]. Observation cameras or direct observations can display the behavior of the animals without disturbance by handling. Behavioral reactions to pain include specific pain-associated behaviors—such as hunched back posture, tail wagging and rump scratching—and unspecific changes in physiological behaviors such as suckling or sleeping [8]. However, observation of animal behavior over a long period is very labor intensive and requires the experience and training of the observer [7].

More recent behavior-based assessment systems such as grimace scales, which allow classifying pain based on animal mimics, or standardized behavior tests can facilitate and standardize pain assessment because they assess only specific behavior elements and require no comprehensive observation. Regarding pigs, the use of the grimace scale is still challenging, so behavior observation seems to be a more reliable method [7].

There is currently no gold standard for the assessment of pain behavior in piglets [7]. Therefore, to assess the efficacy of local anesthesia in eliminating pain during and after surgical castration, we analyzed the pain behavior of differently treated piglets under eight days of age in an observation arena. Specifically, we compared the acute pain behavior of piglets where anesthesia had been administered and castrated piglets with the behavior of non-castrated piglets or piglets castrated without anesthetic treatment.

## 2. Materials and Methods

The present study is part of the joint project “Effectiveness of local anesthesia in eliminating pain during piglet castration,” a collaboration between the Center for Preclinical Research of the Technical University of Munich and the Clinic for Swine of the Ludwig Maximilian University of Munich. The study consisted of three sequential experimental parts (part 1, part 2 and part 3) designed to find the most effective method and the most suited agent for anesthesia during castration. It complied with all legal requirements regarding the planning and conducting of animal research according to the Directive 63/2010/EU of the European Parliament and of the Council and according to the German Animal Welfare Act (2019). The study was approved by the district government of Upper Bavaria (reference number ROB-55.2-2532.Vet_02-19-11).

### 2.1. Animals and Housing

In total, 178 male hybrid fattening piglets (crossbreed Piétrain × German Landrace and German Large White) were included in the study. The piglets were between three and seven days old (mean ± SD: part 1: 5.1 ± 1.1 days; part 2: 6.6 ± 0.7 days, part 3: 6.7 ± 1.5 days) and had a minimum weight of 1.4 kg. The average weight (mean ± SD) was 2.2 ± 0.5 kg in part 1, 2.2 ± 0.6 kg in part 2 and 2.3 ± 0.6 kg in part 3. Only animals with completely descended testicles and sound general conditions were included in the study and randomly allocated to the different study groups.

All animals were housed on a farm in Bavaria. This breeding and fattening farm housed eighty sows, which farrowed in a three-week rhythm. Sows were fed a mixed ration, including a minimum of 200 g of raw fiber per day, that was produced and processed at the farm. The farrowing pens were 3.8 m^2^ in size (length: 1.65 m, width: 2.30 m) and had partially slatted concrete floors that were covered with rubber mats in the lying area of the sow. The piglet creep had a solid concrete floor bedded with sawdust and straw. A movable cover and an infrared heat lamp (heat lamp with protective case, 250 W, ARTAS, Armstadt, Germany) were installed above the piglet creep. To avoid pain and thus a possible influence of additional zootechnical procedures on the assessed parameters, the piglets were ear-tagged after completing the trials and before weaning from the mother sow, and their tails were not docked. On the day before castration, all piglets received an intramuscular iron injection (1 mL Ursoferran^®^, 200 mg active ingredient per milliliter of injection solution for swine, Serumwerk Bernburg AG, Bernburg, Germany).

To allow the collection of behavior data without influence of the observing person, the piglets were placed in an observation arena (modified after Di Giminiani et al. [9]); Figure 1 at five observation time points (see timeline in Figure 2; for a maximum duration of two minutes per observation). This arena consisted of a round black plastic tub with a diameter of 60 cm at the bottom and 70 cm at the rim, with a height of 40 cm. The tub had four evenly spread side openings of 4 cm diameter per opening, through which the lenses of four digital cameras (GoPro Hero7 Black Action Cam 4K, GoPro Inc., San Mateo, CA, USA) were inserted. The bottom of the arena was covered with a fitted round hemp mat (hemp farrowing mat, 5 mm thick, GFS Top Animal Shop, Ascheberg, Germany) to ensure slip-proof standing of the piglets and to absorb excrement. The hemp mat was replaced as needed, e.g., when soiled. To keep the piglet warm and provide adequate lighting, a heat lamp (heat lamp 250 W, ARTAS, Armstadt, Germany) was placed 70 cm above the bottom of the observation arena. The arena and the metal frame were mounted on a wooden platform (1 × 1 m) so that the whole construction was easy to lift and move. During the recordings, the observation arena was placed in the aisle next to the farrowing pens so that the piglets could always hear and smell their mother sow and litter siblings.

### 2.2. Experimental Design

The study was divided into three parts that were established step by step to optimize the applied local anesthesia. In part 1, 4 local anesthetics were tested on 71 piglets from 16 litters. In part 2, 2 injection techniques were applied with 1 selected local anesthetic on 47 piglets from 10 litters. In part 3, the most effective local anesthetics and injection techniques identified in the first two parts were used [10,11,12]. In addition, local anesthesia in part 3 was combined with systemic analgesia. Part 3 was conducted on 60 piglets from 11 litters. In addition to the treatment groups, each study part included two control groups: a “handling” group in which injection and castration were only simulated (sham group) and a sodium chloride (NaCl) group in which the piglets were injected with NaCl solution instead of local anesthetic and thus were castrated without anesthesia. Table 1 gives an overview of the study groups investigated in study parts 1–3.

### 2.3. Castration

On the day before castration, the piglets were weighed, and their health status was examined. To ensure individual identification of the piglets, they were marked with distinct symbols on the back by using a paintbrush and black marker spray (MS marker spray, black, MS Schippers, Kerken, Germany).

Before castration, the piglets were weighed again on the day of castration with a digital scale (MS Schippers, Kerken, Germany). Afterwards, the piglets were individually filmed in the observation arena for a maximum duration of two minutes to establish a baseline of the behavior. This baseline could then be compared to the recordings after the procedure. For the injection, the piglets were fixed in a castration stand (MS Schippers, Kerken, Germany) and injected into both testicles with either a local anesthetic or NaCl solution by means of a self-filling injector (HSW ECO-MATIC^®^, Henke-Sass, Wolf GmbH, Tuttlingen, Germany) fitted with a 25-gauge needle (0.5 × 16 mm, B. Braun TravaCare GmbH, Hallbergmoos, Germany). In part 1 of the study, the injection was carried out in two steps (two-step injection), during which the left and right testicles were first injected with 0.5 mL injection solution administered intratesticularly and then with 0.5 mL administered subcutaneously into the scrotum. Calculated for the mean weight of 2.2 kg, piglets in part 1 of the study received 36.4 mg/kg Procaine (Pronestesic 40 mg/mL injection solution for horses, cattle, swine and sheep; FATRO S.p.A, Ozzano Emilia (Bologna), Italy), 18.2 mg/kg Lidocaine (Xylocitin^®^ 2% with epinephrine 0.001%; mibe GmbH Arzneimittel, Brehna, Germany), 4.5 mg/kg Bupivacaine (Bupivacaine 0.5%, JENAPHARM^®^; mibe GmbH Arzneimittel, Brehna, Germany) and 18.2 mg/kg Mepivacaine (Mepidor^®^ 20 mg/mL injection solution for horses; Wirtschaftsgenossenschaft deutscher Tierärzte eG, Garbsen, Germany). Based on a dosage adjustment, less local anesthetic was injected in part 2 of the study (calculated for the mean weight of 2.2 kg: 10.9 mg/kg Lidocaine). In part 3 of the study piglets received 10.4 mg/kg Lidocaine (calculated for the mean weight of 2.3 kg) and 12.2 mg/kg Mepivacaine. The two-step injection technique according to Hansson et al. [13] (two-step [H] injection) used in parts 2 and 3 started with an intratesticular injection of 0.4 mL per testicle, and the remaining 0.2 mL was delivered during withdrawal of the needle instead of a subcutaneous injection into the scrotum. The piglets of the sham group (handling only) were also fixed in the castration stand, and an injection was simulated by pressure exerted with the needle cap while the testicles were fixed with the fingers. The one-step fenestrated injection technique (one-step [F] injection) used for comparison in parts 2 and 3 was carried out with a 25-gauge needle (0.5 × 10 mm) that had four lateral openings in addition to the distal opening, two of them proximally and two distally [12]. Thus, during injection, the local anesthetic was simultaneously administered distally into the testicle and proximally into the subcutis of the scrotum. Afterwards, the piglets were filmed in the observation arena for maximally two minutes and then placed back into the farrowing pen with their mother sow.

To ensure full activity of the administered local anesthetics, the piglets were castrated in the castration stand after a waiting period of 20 min. After cleaning the scrotal area with antiseptic solution (Octenisept^®^, Schuelke & Mayr GmbH, Norderstedt, Germany), the piglets were castrated with a sterile surgical blade (scalpel blades carbon steel, sterile 21, Heinz Herenz Medizinalbedarf GmbH, Hamburg, Germany; scalpel handle no. 4, AESCULAP AG & CO. KG, Tuttlingen, Germany). First, the skin and the tunica vaginalis were incised on one side. Then, the testicle was exposed and removed after severing the ductus spermaticus. This procedure was repeated on the other side. The piglets of the sham group were fixed in the castration stand, and the castration was simulated by manual fixation of the testicles and pressure exerted with the dull end of the scalpel. Castration and injection were standardized and always conducted by the same three trained and alternating veterinarians. Immediately after castration, the piglets were placed in the observation arena for a maximum of two minutes. Two hours after castration, all piglets were again observed in the observation arena for maximally two minutes.

On the day after castration (approximately 24 h after castration), the weight and general condition of the piglets were re-assessed, and all piglets were once more filmed in the observation arena for maximally two minutes. Each piglet then received 0.4 mg/kg of an analgesic (Metacam^®^ 5 mg/mL, injection solution for cattle and swine, Boehringer Ingelheim Pharma GmbH & Co. KG, Ingelheim am Rhein, Germany), whose administration had been delayed until this time to exclude the influence of medication on the data collected in the observation arena. An exception was made for the piglets of study part 3; immediately before injection or simulated injection, these piglets received systemic, intramuscularly administered analgesics: 0.4 g/kg meloxicam (Metacam^®^ 5 mg/mL, injection solution for cattle and swine, Boehringer Ingelheim Pharma GmbH & Co. KG, Ingelheim am Rhein, Germany) and 50 mg/kg metamizole (Metamizol WDT 500 mg/mL, injection solution for horses, cattle, swine and dogs, Wirtschaftsgenossenschaft deutscher Tierärzte eG, Garbsen, Germany).

### 2.4. Behavior in the Observation Arena

Each piglet was placed in the observation arena for a maximum duration of two minutes at each of the following five time points: (i) baseline, i.e., on the day of castration before all applied procedures; (ii) after injection or sham injection; (iii) after castration or sham castration; (iv) two hours after castration or sham castration; (v) and twenty-four hours after castration or sham castration. According to predetermined termination criteria, piglets were removed from the observation arena if they showed signs of stress or fear during direct observation. These criteria included stress reactions such as freezing, high-frequency vocalization, flight attempts, or frequent pivoting (see Appendix A). The recorded behavior was analyzed by focal animal sampling, according to Martin and Bateson [14] for up to 60 s. For the analysis, an ethogram modified after Hay et al. [8] was used (Table 2). All observers were blinded regarding the study group.

### 2.5. Statistical Analysis

For an initial description of the data, relative frequencies were computed for all behavior variables along study parts, study groups and time points. For further analysis, generalized linear mixed models for binary response variables with logit link function were used for each part independently to estimate risks for the different behaviors along study groups and time points. All models included study group and time point as fixed effects, the interaction of these variables, and the weight of the animals. Subject-specific variation was accounted for by random effects for the intercept. Comparisons of estimated risks between study groups at given time points were performed by risk ratios. Results are presented as estimated risks, risk ratios, the corresponding 95% uncertainty intervals and *p*-values. For analysis, the R language for statistical computing [15] was used. Inter-observer reliability was calculated with Cohen’s kappa from the agreement between two observers.

## 3. Results

The most frequently shown pain-associated behaviors were changes in tail position, hunched back posture, and hunched back posture with pressing (against the arena wall), confirming previously published findings [8,16,17]. In all study parts, we found at least one significant increase in these parameters between the baseline behavior and the behavior at subsequent time points. We also found differences between the subsequent time points in that the frequency of the behavior reached a peak at castration (changes in tail position, tail wagging) or two hours after castration (hunched back posture, hunched back posture with pressing against the arena wall) and decreased afterwards. In the sham group, the parameters were often only increased at single time points. The intraclass correlation coefficient (Cohen’s kappa) was Κ = 0.71 for changes in tail position, Κ = 0.78 for tail wagging, Κ = 0.78 for hunched back posture, and Κ = 0.77 for hunched back posture with pressing against the arena wall.

### 3.1. Study Part 1: Evaluation of Four Local Anesthetics in Comparison with Two Control Groups (Sham; Injection with NaCl, i.e., Castration without Anesthesia)

The piglets of group NaCl Two-Step Part 1 (control group: castration without anesthesia) showed a significant increase in the pain parameter changes in the tail position immediately after castration in comparison with all other groups (*p* = 0.044). Two hours after castration, piglets of group Bupi Two-Step Part 1 showed changes in tail position more frequently than piglets of group Lido Two-Step Part 1 (Figure 3).

Tail wagging after injection occurred more in groups Lido Two-Step Part 1, Mepi Two-Step Part 1 and NaCl Two-Step Part 1 and less in groups Sham Part 1 (control group: handling only) and Proca Two-Step Part 1 (*p* ≤ 0.014). On the day after castration, this behavior was shown more frequently in group Bupi Two-Step Part 1 than in all other groups (*p* ≤ 0.016), especially in comparison with group Sham Part 1 (*p* < 0.001) (Figure 3).

The parameters of hunched back posture and hunched back posture with pressing against the arena wall showed almost no significant differences in this part of the study (Figure 3). We found an increase in hunched back posture in piglets of group Sham Part 1 two hours after castration, with a significant difference to group Mepi Two-Step Part 1 (*p* = 0.046) (Figure 3).

In this part of the study, we found the significant effects of body weight on the behavior parameter of tail wagging. Heavy piglets showed this behavior more frequently than lighter ones (*p* = 0.028).

### 3.2. Study Part 2: Evaluation of Two Injection Techniques for Lidocaine in Comparison with Two Control Groups (Sham; Injection with NaCl, i.e., Castration without Anesthesia)

In study part 2, changes in tail position occurred especially often in groups NaCl One-Step (F) Part 2 and Lido Two-Step (H) Part 2 (Figure 4). Furthermore, two hours after castration the piglets of group NaCl One-Step (F) Part 2 showed a hunched back posture more often than the piglets of group Sham Part 2 (*p* = 0.030) (Figure 4). We found no significant influence of body weight on the frequency of the assessed pain parameters in this part of the study (*p* > 0.50).

### 3.3. Study Part 3: Evaluation of Two Local Anesthetics Combined with Systemic Analgesia in Comparison with Two Control Groups (Sham; Injection with NaCl, i.e., Castration without Anesthesia)

As observed in the preceding study part, immediately after castration the piglets of group Sham Part 3 showed changes in tail position less often than the piglets of the other groups, especially in comparison with group Lido Two-Step (H) Part 3 (*p* = 0.012). In addition, two hours after castration, the piglets of all other groups showed this behavior more often than the piglets of group Sham Part 3 (*p* < 0.001) (Figure 5). As found in part 2, two hours after castration, the frequency of hunched back posture was lower in group Sham Part 3 than in group NaCl One-Step (F) Part 3 (*p* = 0.036) (Figure 5).

In this part of the study, we found an influence of body weight on one pain parameter. The possibility for the occurrence of a hunched back posture with pressing against the arena wall increased with increasing body weight (*p* = 0.008).

## 4. Discussion

The first two publications of the project team presented results of (i) physiological parameters assessed in piglets castrated under low doses of isoflurane and local anesthesia according to a minimal anesthesia model [10] and (ii) defensive movements, vocalization and the behavior in a handling chute assessed in conscious piglets under local anesthesia during or after castration [11]. The results showed that castration without anesthesia is painful, confirming previous findings [18]. However, findings on the effectiveness of local anesthesia during castration have been diverging. In the present study, the occurrence of specific acute behavioral signs of pain could be reduced by the administration of local anesthetics. This result agrees with those of other publications [13,19].

Because castration-related pain induces behavior changes that can persist for up to five days [8,20], we assessed the pain behavior 2 and 24 h after castration based on video recordings made in an observation arena. Changes in tail position were previously described to be associated with various emotions in pigs and piglets [21,22,23]. The association between specific tail positions and pain could be demonstrated multiple times [21,24,25]. In our study, this behavior occurred most often immediately after castration. Moreover, in comparison with the behavior before the procedure, changes in tail position were still markedly increased 2 or 24 h after castration. Furthermore, a change in tail position was shown significantly more often by castrated than by sham-castrated piglets, suggesting a pain-induced expression of this behavior.

Several studies focused more on the movement than on the position of the tail [8,16,26,27] and mostly found an increase in tail movements after castration. In the present study, we assessed tail movements and tail position separately and found an increase in tail movements (tail wagging) rather on the day after castration or after injection of the local anesthetic. This parameter could be associated with paresthesia—such as itching (due to increased bleeding) or numbness (after administration of the local anesthetic). Other studies also reported increased tail wagging after castration with local anesthesia, whereas piglets castrated without anesthesia showed less tail wagging [27]. The considerably high occurrence of tail movements in the piglets of group Bupi Two-Step Part 1 on the day after castration could be caused by the long-lasting effect of bupivacaine, which induces numbness [28]. Thus, we could not demonstrate the pain-specificity of the tail-wagging parameter.

The hunched-back posture parameter has been classified as a pain parameter for various animal species, e.g., for cattle [29]. This behavior was previously observed in piglets after castration and persisted up to several hours after the procedure [17,30]. In our study, we found a short-term increase in this parameter also in the sham-castrated piglets—especially in study part 1 and considerably less after preemptive systemic pain medication in study part 3. We assume that fixation of the testicles during handling can already induce pain stimuli that cause a piglet to change its body position and hunch its back. We also found an increase in the hunched-back posture parameter by pressing against the arena wall in the sham-castrated piglets at the time of castration. To ensure an entirely pain-free control group, future assessments in the observation arena should include a control group without handling.

In general, we found differences between castrated and sham-castrated piglets for only a few of the assessed parameters. These results are in line with other studies comparing the behavior of castrated and sham-castrated piglets before and after castration [7]. Differences between the different local anesthetics or injection techniques were even less pronounced. It is possible that a larger number of animals might reveal clearer differences. However, in the NaCl group and in the groups treated with local anesthetics, we noticed a sustained increase in the behavioral pain parameters, which persisted until the day after castration, even with the preemptive administration of meloxicam and metamizole. Various studies have confirmed that the pain induced by castration seems to persist for several days and that swellings and redness can remain visible and palpable for several days [8,11,27]. Therefore, perioperative pain management should go beyond treating acute castration-induced pain. Several review articles emphasized the relevance of observing specific pain behavior [7,31]. However, continuous observation is associated with high technological effort and labor input. Some researchers, therefore, used short observation time points to conduct behavior observations on piglets [9]. Observation in short time intervals (focal sampling) is suited for the assessment of highly frequent specific behaviors, whereas rarely shown behavior parameters are easy to miss with this method. Furthermore, the use of an arena like the one in our study only allows the evaluation of individual behavior. Social behaviors—such as play behavior—cannot be assessed.

## 5. Conclusions

To evaluate effective anesthesia in piglets undergoing castration, most studies focus on the acute pain at the time or on the day of castration. However, castration-induced pain persists over a longer period. Thus, for animal welfare reasons, the associated long-lasting pain should be included in the evaluation of a castration method. Our results suggest that the administration of local anesthetics seems to reduce pain during castration but cannot eliminate the associated longer-lasting pain, owing to the limited duration of action of the local anesthetics. The recommendation of a specific local anesthetic or injection technique is not possible by means of the presented behavior analysis in the observation arena. However, it was noticeable that some of the local anesthetics—such as lidocaine and mepivacaine—led to stronger or longer-lasting decreases in the pain-associated behavior parameters. The “changes in tail position,” “hunched back posture” and “hunched back posture with pressing against the arena wall” parameters established in this study seem to allow a valid assessment of pain in piglets and generated reproducible results facilitating differentiation between the two control groups (NaCl (castration without anesthesia) and Sham (no castration, only handling)). Because pain-induced behavioral changes in piglets are individual and transient, further studies are needed to evaluate how far pain indicators differ according to the age and race of the assessed piglets. In the long term, a standardized assessment protocol for pain detection in piglets would be useful to allow better comparability between studies and better evaluation of the suitability of alternative methods of anesthetization.

## Figures and Tables

**Figure 1 animals-13-00529-f001:**
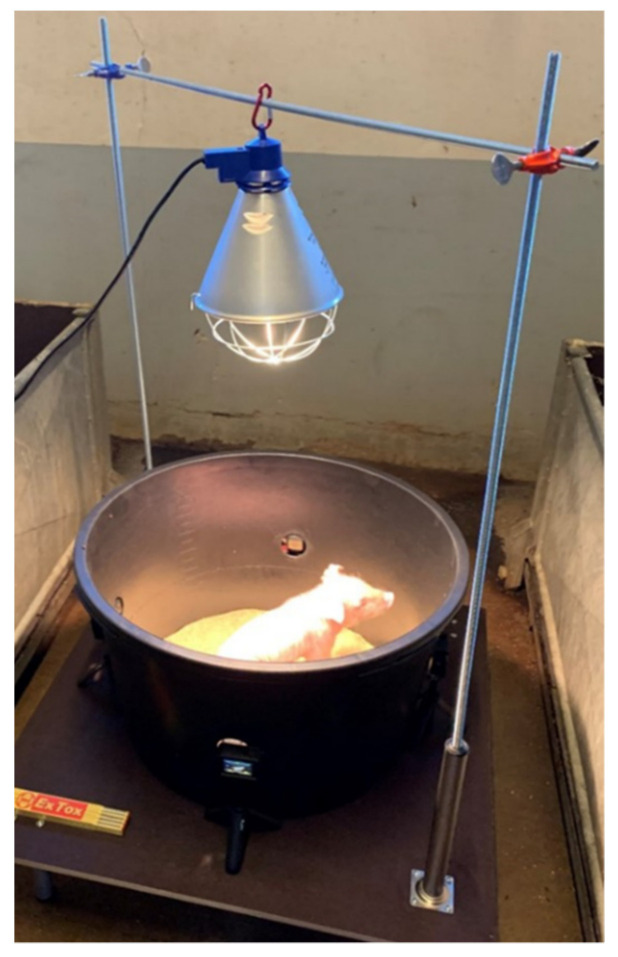
Photograph of the observation arena with a standing piglet, modified after Di Giminiani et al. [9]. Photo credit: S.Bergmann.

**Figure 2 animals-13-00529-f002:**
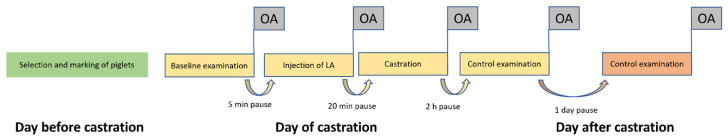
Experimental procedure from the day before castration until the day after castration showing all five observation time points in the observation arena. LA = Local anesthetic; OA = video recording in the observation arena.

**Figure 3 animals-13-00529-f003:**
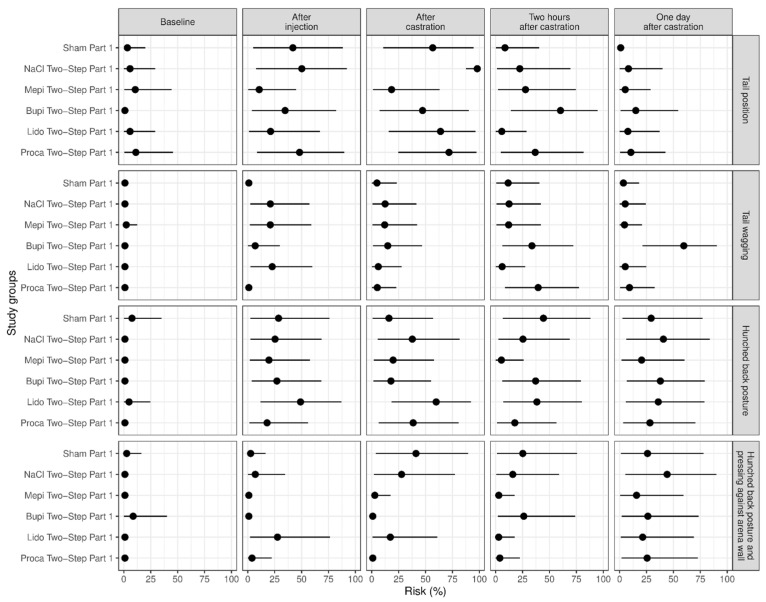
Study part 1: Effect of the study group on the relative risk for the percentage occurrence (median with minimum and maximum) of the parameters changes in tail position, tail wagging, hunched back posture and hunched back posture with pressing against the arena wall. The different study groups were compared according to the five assessed time points baseline (before any procedure), after injection, after castration, two hours after castration and one day after castration. The study groups were Sham Part 1 (n = 11), NaCl Two-Step Part 1 (n = 12), Mepi Two-Step Part 1 (n = 12), Bupi Two-Step Part 1 (n = 12), Lido Two-Step Part 1 (n = 12) and Proca Two-Step Part 1 (n = 12).

**Figure 4 animals-13-00529-f004:**
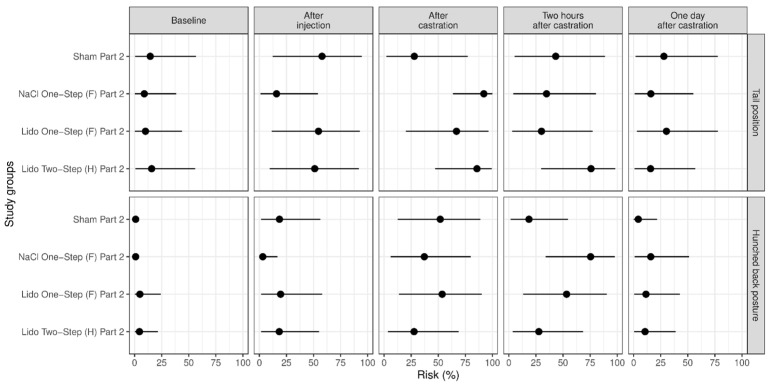
Study part 2: Effect of the study group on the relative risk for the percentage occurrence (median with minimum and maximum) of the parameter changes in tail position and hunched back posture. The different study groups were compared according to the five assessed time points baseline (before any procedure), after injection, after castration, two hours after castration and one day after castration. The study groups were Sham Part 2 (n = 12), NaCl One-Step (F) Part 2 (n = 12), Lido Two-Step (H) Part 2 (n = 12) and Lido One-Step (F) Part 2 (n = 11).

**Figure 5 animals-13-00529-f005:**
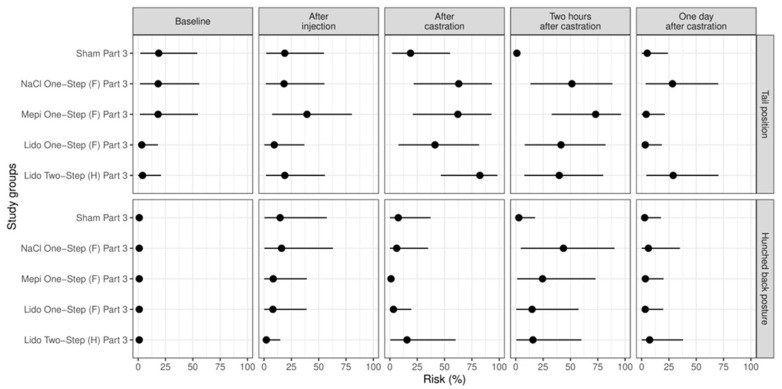
Study part 3: Effect of the study group on the relative risk for the percentage occurrence (median with minimum and maximum) of the parameter changes in tail position and hunched back posture. The different study groups were compared according to the five assessed time points baseline (before any procedure), after injection, after castration, two hours after castration and one day after castration. The study groups were Sham Part 3 (n = 12), NaCl One-Step (F) Part 3 (n = 12), Mepi One-Step (F) Part 3 (n = 12), Lido Two-Step (H) Part 3 (n = 12) and Lido One-Step (F) Part 3 (n = 12).

**Table 1 animals-13-00529-t001:** Study groups administered drugs and injection techniques in study parts 1–3.

Study Part	Study Group	Drug	Product Information and Brand Name	Dosage	Injection Technique	Number of Piglets
1	Proca Two-Step Part 1	Procaine hydrochloride + adrenaline	Pronestesic 40 mg/mL injection solution for horses, cattle, swine and sheep; FATRO S.p.A, Ozzano Emilia (Bologna), Italy	0.5 mL i.t. 0.5 mL s.c.	Two-step technique	12
	Lido Two-Step Part 1	Lidocaine hydrochloride + adrenaline	Xylocitin^®^ 2% with epinephrine 0.001%; mibe GmbH Arzneimittel, Brehna, Germany	0.5 mL i.t. 0.5 mL s.c.	Two-step technique	12
	Bupi Two-Step Part 1	Bupivacaine hydrochloride + adrenaline	Bupivacaine 0.5%, JENAPHARM^®^; mibe GmbH Arzneimittel, Brehna, Germany	0.5 mL i.t. 0.5 mL s.c.	Two-step technique	12
	Mepi Two-Step Part 1	Mepivacaine hydrochloride	Mepidor^®^ 20 mg/mL injection solution for horses; Wirtschaftsgenossenschaft deutscher Tierärzte eG, Garbsen, Germany	0.5 mL i.t. 0.5 mL s.c.	Two-step technique	12
	Sham Part 1	n/a	n/a	n/a	Sham injection	11
	NaCl Two-Step Part 1	Sodium chloride	NaCl 0.9% intravenous infusion; B. Braun Melsungen AG, Melsungen, Germany	0.5 mL i.t. 0.5 mL s.c.	Two-step technique	12
2	Lido One-Step (F) Part 2	Lidocaine hydrochloride + adrenaline	Xylocitin^®^ 2% with epinephrine 0.001%; mibe GmbH Arzneimittel, Brehna, Germany	0.6 mL	One-step, fenestrated needle	11
	Lido Two-Step (H) Part 2	Lidocaine hydrochloride + adrenaline	Xylocitin^®^ 2% with epinephrine 0.001%; mibe GmbH Arzneimittel, Brehna, Germany	0.4 mL i.t. 0.2 mL s.c.	Two-step after Hansson et al. 2011	12
	Sham Part 2	n/a	n/a	n/a	Sham injection	12
	NaCl One-Step (F) Part 2	Sodium chloride	NaCl 0.9% intravenous infusion; B. Braun Melsungen AG, Melsungen, Germany	0.6 mL	One-step, fenestrated needle	12
3	Lido One-Step (F) Part 3	Lidocaine hydrochloride + adrenaline *	Xylocitin^®^ 2% with epinephrine 0.001%; mibe GmbH Arzneimittel, Brehna, Germany	0.6 mL	One-step, fenestrated needle	12
	Lido Two-Step (H) Part 3	Lidocaine hydrochloride + adrenaline *	Xylocitin^®^ 2% with epinephrine 0.001%; mibe GmbH Arzneimittel, Brehna, Germany	0.4 mL i.t. 0.2 mL s.c.	Two-step after Hansson et al. 2011	12
	Mepi One-Step (F) Part 3	Mepivacaine hydrochloride *	Mepidor^®^ 20 mg/mL injection solution for horses; WDT eG, Garbsen, Germany	0.7 mL	One-step, fenestrated needle	12
	Sham Part 3	n/a *	n/a	n/a	Sham injection	12
	NaCl One-Step (F) Part 3	Sodium chloride *	NaCl 0.9% intravenous infusion; B. Braun Melsungen AG, Melsungen, Germany	0.6 mL	One-step, fenestrated needle	12

Abbreviations: n/a = not applicable; i.t. = intratesticular; s.c. = subcutaneous; * = piglets were given systemic analgesia prior to the injection of local anesthetic (0.4 g/kg meloxicam (Metacam^®^ 5 mg/mL), injection solution for cattle and swine, Boehringer Ingelheim Pharma GmbH & Co. KG, Ingelheim am Rhein, Germany) and 50 mg/kg metamizole (Metamizol WDT 500 mg/mL, injection solution for horses, cattle, swine and dogs, Wirtschaftsgenossenschaft deutscher Tierärzte eG, Garbsen, Germany).

**Table 2 animals-13-00529-t002:** Ethogram for pain-associated behaviors in the observation arena, modified after Hay et al. [8].

Functional Circle	Behavior	Definition
Pain behavior	Tail position	Tail held straight and not curled, either extended from the rump or hanging down
	Tail wagging	Repeated tail movements up and down or side to side
	Hunched back posture	Kyphosis, front legs extended
	Hunched back posture with pressing against the arena wall	Kyphosis, front legs extended, rump pressed against the wall of the observation arena
	Sitting	Body weight supported by hind quarters and front legs
Excretion behavior	Hunched back posture during excretion	Kyphosis during excretion of feces

## Data Availability

Not applicable.

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
