# Peer review of "Behavior of Piglets in an Observation Arena before and after Surgical Castration with Local Anesthesia"

_animals, 2023, doi:10.3390/ani13030529_

Round 1
Reviewer 1 Report
Please see attached word document.

Author Response
Please see attached pdf document.

Reviewer 2 Report
I have been fascinated to read this manuscript and would like to commend the authors for a solid body of research focused on a serious welfare concern for a huge number of piglets each year.
I have just a few minor comments.
Table 1
Why was a larger volume of mepivacaine used (0.7ml) vs 0.6ml for lidocaine?
Should this table also state that systemic analgesics (meloxicam and metamizole) were given before castration for part 3?
Discussion
This is very thorough, including work on piglet grimace scales and mood assessment. I just wonder whether some of the more wearable ‘tech’ modalities of more objective pain assessment (heart rate variability and electrodermal activity [which I think may be used in the PainTrace device??) could be used alongside behavioural assessments, perhaps to help validate the latter in your future studies?
References
Reference 32 seems to be superfluous?
Surgical castration method:- Could I just check – Am I right that the spermatic cord was transected in this study (not ligated or tensed until it snapped)?
I was just thinking that traction on the spermatic cord (even to expose the testicle), also increases tension on the nerves it contains, many of which have a lumbar origin, so I wonder if this could contribute to a kyphotic stance (indicative of ‘back pain’) post castration. But also I seem to remember reading that castration involving traction on the spermatic cord until it snaps, may result in breakage of the spermatic artery near the aorta and possible haemorrhage – which may result in ‘back pain’ after castration… but I cannot find the article.
Discussion
Intratesticular lidocaine appears to be systemically absorbed very quickly, and, at least in horses, it has been proposed that castration be performed within 10 minutes of intratesticular injection (Haga et al. 2006). The same author also castrated piglets 10min after intratesticular and intrafunicular lidocaine (Haga and Ranheim 2005). I wonder if 20 minutes, as used in your study, was when the local anaesthetic effect was subsiding which might explain some of the variability in your results?
Haga HA et al. (2006) Effect of intratesticular injection of lidocaine on cardiovascular responses to castration in isoflurane-anesthetized stallions. Am J Vet Res 67, 403-8.
Haga HA, Ranheim B (2005) Castration of piglets: the analgesic effects of intratesticular and intrafunicular lidocaine injection. Vet Anaesth Analg 32, 1-9.
Ranheim B et al. (2005) Distribution of radioactive lidocaine injected into the testes in piglets. J Vet Pharmacol Ther 28, 481-3.
Author Response
Dear reviewer,
thank you for your favorable review of our manuscript.
We would like to address your comments as follows:
- Dosage of mepivacaine and lidocaine:
Our study was part of a bigger project to research the effectiveness of local anesthesia during piglet castration. Additionally, to the study parts referred to in this paper, we conducted research on piglets under light isoflurane anesthesia (for further information, please refer to: https://doi.org/10.3390/ani12081028). Based on the isoflurane trials the dosage of the local anesthetics was continuously refined. Thus, we elected to use mepivacaine in a higher dosage of 0.7ml in Part 3 of this study.
- Table 1 Inclusion of systemic analgesia into Table 1: We have included the dosage and drug names of the systemic analgesia in a footnote.
- We acknowledge that the combination of different methods for pain assessment give the most reliable results in piglets. In this study several assessment methods have been used in addition to behavioral assessments, including defensive movements, vocalization, and navigation time of a handling chute. Some of these findings have been published already (please refer to: https://doi.org/10.3390/ani10101752 and https://doi.org/10.3390/ani12182309 ) or will be published in the future.
We tried to influence the piglets as little as possible during their time in the observation arena because we had seen during the training phase that even small noises or movement by observers affected the piglets’ behavior.
We discussed the possibility of additional thermographic imaging in future studies to have another objective parameter for pain assessment. This could be included without requiring any additional handling of the piglets.
- Reference 32:
Thank you for pointing out that reference as superfluous. We have removed this reference in the edited version.
- In our study we transected the spermatic cord by cutting it with a scalpel. As the sham castrated piglets showed an increase in kyphotic stance as well, we assumed that the manipulation of the testes when they are held in position before the cutting of the skin (or simulation of cutting) might already be a cause of visceral painfulness.
- Waiting period of 20 minutes after injection of local anesthetic:
For lidocaine the absorption happens indeed rather quickly after just 5 to 10 minutes. However, the effect is expected to last for at least 2 to 3 hours if combined with epinephrine (Eva Rioja Garcia in Veterinary Anesthesia and Analgesia: The Fifth Edition of Lumb and Jones.
Edited by Kurt A. Grimm, Leigh A. Lamont, William J. Tranquilli, Stephen A. Greene and Sheilah A. Robertson. © 2015 John Wiley & Sons, Inc. Published 2015 by John Wiley & Sons, Inc.p.343).
As several different local anesthetics were used in the first part of the study the waiting period was chosen to be 20 minutes for all to ensure all used agents had been properly absorbed. In the subsequent study parts, we kept the waiting period at the same length to ensure comparability of results.